# Role of Pretransplant Treatments for Patients with Hepatocellular Carcinoma Waiting for Liver Transplantation

**DOI:** 10.3390/cancers14020396

**Published:** 2022-01-13

**Authors:** Kohei Ogawa, Yasutsugu Takada

**Affiliations:** Department of HBP and Breast Surgery, Ehime University, Shitsukawa, Toon 791-0295, Ehime, Japan; takaday@m.ehime-u.ac.jp

**Keywords:** liver transplantation, hepatocellular carcinoma, bridging, downstaging, locoregional therapy

## Abstract

**Simple Summary:**

Hepatocellular carcinoma (HCC) is the fifth most common cancer in men worldwide and the second leading cause of cancer death. Liver transplantation (LT) is one of the treatment options for patients with HCC. Recently, there have been many reports of the usefulness of locoregional therapy, such as transarterial chemoembolization and radiofrequency ablation, for HCC as pretreatment before LT. In Western countries, locoregional therapy is used to bridge until transplantation to prevent drop-outs from the waiting list or for downstaging to treat patients with advanced HCC who initially exceed the criteria for LT. With the progress of locoregional therapy, new reports on the effects of bridging and downstaging locoregional therapy as pretransplant treatment are increasing in number.

**Abstract:**

Recently, there have been many reports of the usefulness of locoregional therapy such as transarterial chemoembolization and radiofrequency ablation for hepatocellular carcinoma (HCC) as pretreatment before liver transplantation (LT). Locoregional therapy is performed with curative intent in Japan, where living donor LT constitutes the majority of LT due to the critical shortage of deceased donors. However, in Western countries, where deceased donor LT is the main procedure, LT is indicated for early-stage HCC regardless of liver functional reserve, and locoregional therapy is used for bridging until transplantation to prevent drop-outs from the waiting list or for downstaging to treat patients with advanced HCC who initially exceed the criteria for LT. There are many reports of the effect of bridging and downstaging locoregional therapy before LT, and its indications and efficacy are becoming clear. Responses to locoregional therapy, such as changes in tumor markers, the avidity of FDG-PET, etc., are considered useful for successful bridging and downstaging. In this review, the effects of bridging and downstaging locoregional therapy as a pretransplant treatment on the results of transplantation are clarified, focusing on recent reports.

## 1. Introduction

As of 2018, hepatocellular carcinoma (HCC) is the fifth most common cancer in men worldwide and the second leading cause of cancer death [1]. Liver transplantation (LT) is one of the treatment options in patients with HCC. Since LT can treat not only HCC, but also underlying liver cirrhosis, it is an ideal treatment method for patients with decompensated cirrhosis with HCC who cannot be treated with treatments other than LT. In Europe and the United States (USA), LT is indicated for early-stage HCC regardless of liver functional reserve. In Japan, where there are few deceased donors available, however, locoregional treatment such as liver resection, transarterial chemoembolization (TACE), and radiofrequency ablation (RFA) is indicated while liver functional reserve is good, and LT is selected when liver functional reserve becomes poor, and there is no treatment option other than LT. Therefore, in Europe and the USA, the purpose of pretransplant locoregional therapy is for bridging to LT, preventing patients with early-stage HCC from dropping out of the waiting list, or for downstaging intermediate stage HCC to put the patients on the waiting list. On the other hand, in Japan, locoregional therapy is often given for the purpose of radical cure, and salvage LT is indicated for patients whose liver functional reserve deteriorates gradually with repeated locoregional therapies.

In patients with early-stage HCC, locoregional therapy may shrink the tumor and provide better transplant outcomes, while repeated locoregional therapies may advance HCC, which might result in worsening post-transplant outcomes. On the other hand, in patients with advanced HCC, if downstaging to within the indication criteria for LT is performed successfully by locoregional therapy, the results after LT might be close to those of patients within the indication criteria initially.

This review focuses on the recent literature and clarifies the role of locoregional therapy before LT.

## 2. Indications for Liver Transplantation in Patients with Hepatocellular Carcinoma

In the history of deceased donor LT (DDLT) in Europe and the USA, LT has been performed as a treatment method for unresectable HCC from the early period. However, the results of LT for HCC were dismal until the early 1990s. O’Grady et al. [2] reported that the 2-year survival rate after LT for HCC was 40% or less, and 65% of cases that survived more than 3 months after LT recurred. Ringe et al. [3] reported that the 5-year survival rate was less than 20%. Due to the high recurrence rate, LT for patients with HCC using limited organs has been controversial. On the other hand, as a promising result, Iwatsuki et al. compared the results of LT to hepatectomy according to the TNM classification and reported that the 5-year survival rate of LT in Stage II was 68%, which was significantly better than the 43% of hepatectomy [4]. Furthermore, in 1993, Bismuth et al. [5] reported a 3-year recurrence-free survival rate of 83% after LT in cases with 1–2 lesions ≤3 cm, which was significantly better than that of 44% in cases with ≥3 lesions ≥3 cm and that of 18% after hepatectomy. They showed that tumor size and number affected transplant outcomes, as well as the superiority of LT over hepatectomy. Then, in 1996, Mazzaferro et al. [6] reported that the 4-year survival rate after LT of single HCC ≤5 cm or 1–3 lesions ≤3 cm was 85% (4-year recurrence-free survival rate 92%). Since then, these Milan criteria (single lesion ≤5 cm or 1–3 lesions ≤3 cm) have been used worldwide as indication criteria for DDLT.

Currently, according to the European Association for the Study of the Liver (EASL) clinical practice guideline, HCC within the Milan criteria is divided into very early stage (single, ≤2 cm) and early stage (single ≥2 cm, ≤5 cm, or 2–3 lesions ≤3 cm). If liver function is good, ablation or resection is recommended for HCC in the very early stage, and LT is indicated for patients with HCC in the early stage, for whom liver resection is unsuitable due to portal hypertension [7].

In the USA, organ transplantation is regulated by the United Network for Organ Sharing (UNOS), and HCC patients with decompensated cirrhosis are prioritized by the Model for End-stage Liver Disease (MELD) score. On the other hand, patients with compensatory cirrhosis have a low MELD score, and they do not get to LT by the MELD score. Therefore, priority is determined according to the number and size of HCC and the registration period. HCC within the Milan criteria is divided into stage T1 (EASL very early stage) and stage T2 (EASL early stage), and patients with T2 HCC are prioritized. Since 2005, T2 patients have been given a MELD-Na score of 22 points, and 3 points have been added every 3 months [8]. However, in 2015, the system was revised so that T2 patients are not given the exceptional MELD-Na score until a 6-month mandatory period after enrollment, and a MELD-Na score of 28 points is given after 6 months if the tumor meets the Milan criteria with alpha-fetoprotein (AFP) below a certain level [9]. On the other hand, in Japan, LT is indicated for patients with HCC within the Milan criteria or 5-5-500 criteria (1–5 lesions ≤5 cm, AFP ≤500 ng/mL) with Child-Pugh classification C liver cirrhosis [10]. The priority goes up with a worsening MELD score. For patients with low MELD scores, however, a MELD score of 16 points is given, and 2 points are added every 3 months if HCC meets the criteria. Using this allocation system, even in the USA, where the number of deceased donors is overwhelmingly larger than in Japan, it has been reported that 12% of HCC patients on the waiting list drop-out due to HCC progression in 6 months and 15–30% in a year, and the waiting period is often prolonged more than one year [11,12]. Therefore, in order to get to LT, it is necessary to wait while controlling the progression of HCC.

On the other hand, in living-donor liver transplantation (LDLT), which is not affected by the allocation system, the Milan criteria are considered to be too strict. Therefore, expanded criteria with higher number and size, as well as biological surrogates, such as AFP, des-gamma-carboxyprothrombin (DCP), FDG-PET, and response to pretransplant locoregional therapy, have been reported with results to comparable the Milan criteria [10,13,14,15,16,17,18]. In recent years, models for predicting the recurrence rate of HCC after LT with more accuracy, using various parameters including tumor size, number, and AFP etc., have been reported [19,20,21,22]. Among them, the formula by Ivanics et al. from Toronto, in which many parameters were analyzed using machine learning, a type of artificial intelligence, has been reported to be more accurate than the AFP score and the Model of Recurrence After Liver Transplant (MORAL) score [22].

As morphological data that determine the indication for LT, classical HCC, which shows enhanced staining in the arterial phase and wash-out in the delayed phase on dynamic CT, has been the target at many facilities so far. With recent advances in diagnostic imaging, however, the American College of Radiology proposed the Liver Imaging Reporting and Data System (LI-RADS) in 2011 with the aim of making diagnostic imaging of HCC more universal and more accurate [23]. In the LI-RADS, liver nodules are classified into 5 subclasses: LR-1 (benign), LR-2 (probably benign), LR-3 (boundary lesion), LR-4 (probably HCC), and LR-5 (HCC). It has been reported that the HCC-specific diagnostic accuracy is 37% for LR-3 and 95% for LR-5 [24,25]. Although LT is indicated for tumors classified as LR-5, the impact of LR-3 and LR-4 tumors on LT results has not been clarified so far.

Recently, Mazzaferro et al. [26] have released metroticket 2.0, an online calculator with tumor diameter, total number, and AFP as parameters, which is useful to predict recurrence rates in individual cases accurately. Centonze et al. [27] from Italy investigated the oncological impact of the LI-RADS classification using this metroticket 2.0. They reported that excluding LR-3 and LR-4 nodules resulted in a significant drop in its accuracy. In the future, it is expected that a more accurate recurrence prediction model will be developed by using these more accurate morphological data.

## 3. Bridging Therapy to Liver Transplantation

### 3.1. Is It Meaningful to Perform Bridging Therapy?

To date, there is controversy regarding performing locoregional bridging therapy (BT) for patients waiting for LT. However, due to the allocation system, the patient will be dropped off the waiting list if the HCC progresses and does not meet the transplant indication criteria. In order to perform LT, it is necessary to keep the HCC stage within the transplant indication criteria. Therefore, TACE and RFA are performed as BT in many facilities. In addition to preventing drop-out from the waiting list mentioned above, BT has the potential to shrink tumors and improve the results of LT and exclude biologically aggressive HCC.

### 3.2. When to Perform Bridging Therapy?

In 1999, Llovet et al. [28] reported that the drop-out rate within 6 months was 23% without BT. After that, Ashoori et al. [29] reported that the drop-out rates at 6 months and 12 months were 2.8% and 5.5%, respectively, with RFA and TACE. Habibollahi et al. [30] reported good results of LT by following patients with T1 HCC without BT and performing BT when the waiting period was extended by another 6 months at the time when HCC progressed to T2. Mehta et al. [31] also reported that T1 tumors were followed up every 3 months, and less than 10% of them progressed beyond T2, which is outside the Milan criteria. For the LT candidate, they recommended that it is better to monitor the progress of the T1 tumor without BT, but once the AFP is elevated to ≥500 ng/mL or the tumor is growing in a short period of time, locoregional therapy should be given as soon as possible. Lai et al. [32] reported that when transplant failure was defined as drop-out from the preoperative waiting list or postoperative recurrence in patients within the Milan criteria, one BT reduced transplant failure by 49%, and up to 3 BTs reduced it by 34%. However, the benefits of bridging disappear with four or more BTs. On the other hand, Tan et al. [33] from Singapore reported that there was no difference in the frequency of drop-outs from the waiting list between the BT group and the non-BT group, but the waiting period tended to be longer in the BT group. They concluded that BT might enable the LT candidate with HCC to wait longer. Although there has been no prospective, randomized, controlled trial to date, there is consensus that locoregional therapy for HCC should be given if the waiting period is 6 months or longer to prevent waiting list drop-out during the waiting period. [34,35].

### 3.3. Effect of Bridging Therapy on Hepatocellular Carcinoma Recurrence after Liver Transplantation

Regarding the relationship between BT and recurrence of HCC after LT, there are many reports showing that there was no difference in LT results between the BT group and the non-BT group for HCC within the Milan criteria, and BT was not effective for HCC within the Milan criteria [36,37]. In a multicenter study in the USA, the 1, 3, and 5-year survival rates of 747 patients without locoregional therapy and 2854 patients with locoregional therapy were 89%, 77%, and 68% vs. 85%, 75%, and 68%, respectively (*p* = 0.490), and the 5-year recurrence rate was 11.2% vs. 10.1% (*p* = 0.474), showing no significant differences [38]. In addition, there is a report from South Korea that, when the tumor stage was T1 or T2 (within the Milan criteria), there was no significant difference in the recurrence rate between the BT group and the non-BT group [39]. On the other hand, Oligane et al. [40] reported that the overall survival rate was higher, and the recurrence rate was lower in the BT group with T2 tumors that received MELD exceptions in a study using the UNOS database in the USA. Renner et al. [41] from Germany reported that patients whose HCC progressed from within the Milan to outside the Milan criteria despite BT had lower overall survival and recurrence-free survival than those controlled within the Milan criteria. On the other hand, Ogawa et al. [42] from Kyoto, Japan, reported that if LT was performed within the Kyoto criteria (≤10 lesions ≤5 cm, DCP ≤400 mAU/mL), results of LT were relatively good. They concluded that it was useful to keep the HCC stage within the expanded criteria by BT for the recurrence of HCC after LT even if it exceeds the Milan criteria. Extended criteria incorporating DCP have been reported mainly from Asian countries, including Japan [15,16], and a recent meta-analysis showed that the frequency of HCC recurrence was 5 times higher in cases with a high level of DCP at LT [43].

It has been reported that when locoregional therapy was effective enough to render the tumor completely necrotic, the results after LT were good [44,45,46]. However, imaging examinations often overestimate the therapeutic effect compared to histopathological findings [47,48,49,50]. Rubinstein et al. [51] reported that 77% of patients had viable tumors even though imaging examinations showed complete response (CR). On the other hand, Xu et al. [46] reported that partially necrotic HCC induced by locoregional therapy was associated with an increased risk of lymphatic metastases. Lai et al. [32] reported that patients who show a poor response to locoregional therapy have a predictably greater risk for pretransplant tumor-related delisting or posttransplant recurrence. Therefore, if the effect of locoregional therapy is insufficient, the risk of recurrence of HCC might increase. Table 1 summarizes the results of BT in selected studies in the past 5 years [33,38,39,52,53,54,55].

## 4. Downstaging of Hepatocellular Carcinoma for Liver Transplantation

### 4.1. Downstaging from Outside the Milan Criteria to within the Milan Criteria

Downstaging is performed for the purpose of reducing tumor volume to within the LT indication criteria for patients whose HCC stage is outside the LT indication criteria. Downstaging HCC to within the LT indication criteria (morphological downstaging) is a means for performing LT, but its true purpose is to reduce the risk of recurrence after LT (biological downstaging) by selecting HCCs with low biological aggressiveness that have few recurrences after LT [56]. Since the allocation criteria in many countries were based on the Milan criteria, there are many reports of downstaging from outside the Milan criteria to within the Milan criteria. Total tumor volume (TTV), AFP, etc., have been reported as predictors for successful downstaging from outside the Milan criteria to within the Milan criteria [57,58]. Murali et al. [59] reported that cases with TTV <200 cm^3^ were successfully downstaged. Regarding AFP, Bova et al. reported that AFP <100 ng/mL after locoregional therapy was a predictor of successful downstaging, and Yao et al. [60] reported that AFP > 1000 ng/mL before treatment was a negative factor for downstaging. It has been reported that the results of LT for patients who succeeded in downstaging within the Milan criteria by locoregional therapy were almost the same as when LT was performed for patients within the Milan criteria in both DDLT and LDLT settings [53,60,61,62,63]. On the other hand, there are some reports of a higher recurrence rate after LT in downstaged cases from outside to within the Milan criteria. Ravaioli et al. [64] from Italy reported that the recurrence rate was 7.6% for those who remained within the Milan criteria, 20.9% for those who were downstaged to the Milan criteria, 31.6% for those who failed downstaging, and 30.4% for those who did not downstage, when the upper limit of the tumor was ≤8 cm for single lesion, ≤5 cm for bifocal lesions, and ≤5 lesions with the maximum diameter of 4 cm with the total tumor diameter ≤12 cm. They also reported that pathological findings showed that the frequency of microvascular invasion and moderate to poorly differentiated HCC was higher in cases with downstaging than in cases originally within the Milan criteria. Affonso et al. [54] from Brazil reported that there were more post-transplant recurrences in downstaging cases from outside to within the Milan criteria compared to bridging cases within the Milan criteria (25% vs. 5.81%, *p* = 0.020). The possible cause of a higher recurrence rate in downstaging cases from outside to within the Milan criteria might be the dissociation of imaging and histopathological findings. Kim et al. [65] reported that the results of LT for cases showing pathological downstaging to T2 (within the Milan criteria) were the same as those within the Milan criteria, whereas the results of LT were poor for cases that were not pathologically downstaged. They listed one or less viable tumors on imaging, viable tumor diameter of ≤1 cm, and AFP ≤20 ng/mL as independent predictors for successful pathological downstaging.

### 4.2. Is Downstaging Necessary for Patients Outside the Milan Criteria but within the Expansion Criteria?

The expansion criteria proposed at each transplant institute often incorporate biomarkers in addition to HCC morphology, and LT results using these expansion criteria are reported to be equivalent to those within the Milan criteria. However, most of them include patients within the Milan criteria, and the recurrence rate after LT is reported to be slightly higher in patients within the expansion criteria but outside the Milan criteria than in patients within the Milan criteria. Kamo et al. [66] reported a 5-year recurrence rate of 22% when LT was performed in patients with intermediate-stage HCC within the Kyoto criteria. This recurrence rate was significantly lower than the 66% 5-year recurrence rate in patients with intermediate-stage HCC outside the Kyoto criteria, but higher than their previous 5-year recurrence rate of 5% in patients within the Milan criteria [67]. In addition, Shimamura et al. [10], who examined the 5-5-500 criteria, showed no significant difference between the Milan criteria and the 5-5-500 criteria for recurrence-free survival, but subgroup analysis showed that patients within both the Milan criteria and 5-5-500 did significantly better than patients outside the Milan criteria but within the 5-5-500 criteria. From these data, it seems that it is meaningful to downstage patients with intermediate HCC to within the Milan criteria by preoperative locoregional therapy, even patients with HCC within the expanded criteria, in order to reduce the risk of recurrence after LT.

### 4.3. Comparison of Liver Transplantation after Downstaging with Other Treatments

There was a randomized, controlled trial conducted by Mazzaferro et al. [68] to examine whether LT after downstaging was superior to other treatments for patients with good liver reserve. They randomly divided patients with successful downstaging into an LT group and a control group that was treated with non-transplant therapy when recurrence occurred. They reported that the LT group did better than the control group in both overall survival and tumor event-free survival. On the other hand, regarding the necessity for LT when complete radical treatment is performed by locoregional therapy, Vitale et al. [56] reported comparable results in cases in which complete biological downstaging could be achieved and maintained by strong locoregional therapy to those in the LT group. They concluded that their data strongly support the current allocation system in Italy in which there is no priority in cases obtaining CR by downstaging.

### 4.4. What Is the Upper Limit of Tumor Burden for Downstaging?

In many past reports, although the stage of HCC before downstaging was outside the Milan criteria, most of the cases met the expansion criteria of each institution and the upper limit of tumor burden for downstaging that allows a low recurrence rate after LT was not clear. Shinha et al. reported that, among patients with HCC within the University of California at San Francisco (UCSF) criteria, 82.4% could be downstaged to the Milan criteria, which was significantly higher than the 64.8% of patients with HCC outside the UCSF criteria [69]. They also reported that recurrence was observed in 3 of 10 cases that were originally outside the UCSF criteria even after downstaging to the Milan criteria. In a study using the UNOS database in the USA, Mehta et al. [70] compared recurrence rates among 3 groups: patients within the Milan criteria; patients who were successfully downstaged to within the Milan criteria from the UNOS downstaging inclusion criteria (single lesion of 5 to 8 cm, or up to 3 lesions of 3 to 5 cm with the sum of tumor diameter ≤8 cm, or 4 to 5 lesions with the sum of tumor diameter ≤8 cm); and patients who were successfully downstaged to within the Milan criteria from outside the UNOS downstaging criteria. The 3-year recurrence rates were significantly lower in patients within the Milan criteria group (6.9% vs. 12.8% vs. 16.7%, respectively), but there was no significant difference in the 3-year survival rates between the Milan criteria group and the UNOS downstaging group (83.2% vs. 79.1% vs. 71.4%, respectively). They suggested that the dissociation of preoperative imaging and histopathological findings might be the cause of the higher recurrence rate in the UNOS DS group than in the Milan criteria group. Wu et al. [71] from Taiwan reported that the recurrence rate after LDLT was significantly higher in cases that exceeded the UCSF criteria even with downstaging compared to the cases with or without locoregional therapy within the UCSF criteria. Toso et al. [72] from Switzerland reported that, although the recurrence rate was slightly higher in the downstaging group, there was no difference in survival between the downstaging group and the control group (patients who remained inside the LT criteria) when the tumor burden was limited to TTV ≤115 cm^3^ and AFP ≤400 ng/mL. From these reports, if HCC was pathologically downstaged to within the Milan criteria, the recurrence rate was equivalent to that of the cases within the Milan criteria. However, since dissociation between the imaging findings and the histopathological findings seems to increase with the progress of HCC, it seems reasonable to limit the upper limit of tumor burden for downstaging to cases within the expansion criteria of each institution. On the other hand, in recent years, there have been increasing reports of LT for patients with advanced HCC with portal vein tumor thrombus (PVTT), and among them, Soin et al. [73] from India reported that the transplant results were acceptable in cases with PVTT that were successfully downstaged. In the future, the upper limit of tumors that can be biologically downstaged may be expanded with the progress of preoperative locoregional therapy. Table 2 summarizes the results of downstaging in selected studies of the past 5 years [58,62,63,64,70,72].

## 5. Variations of Pre-Transplant Locoregional Therapies

Although there are many types of pre-transplant locoregional therapies such as hepatic resection, ablation, and TACE, it is not clear which is best, and the application of each therapy, considering its cost-effectiveness, is essential. RFA is less expensive than TACE [74,75] and is more effective in terms of tumor necrosis. However, its use is limited by the size, number, and location of tumors. Although TACE is the most commonly used pre-transplant treatment, it has the disadvantages of adverse effects due to toxicity and higher cost caused by anti-cancer drugs compared to transarterial embolization (TAE). A recent meta-analysis comparing the therapeutic effects of TACE and TAE demonstrated the non-superiority of TACE [76]. TACE is reported to have a tumor response in approximately 60% of cases [37,45,77,78,79,80,81], and the frequency of complete tumor necrosis in excised specimens after LT was 16.7% to 44% [82,83,84]. Tumor response to RFA is reported to be slightly higher than with TACE [85,86,87]. The drop-out rate from the waiting list varies from 3–35% with TACE, and it is around 25% with RFA [45,78,85,86,88,89].

Regarding complications related to locoregional therapies, a recent systematic review showed that TACE increased the risk of hepatic artery-related complications after LT [79,90,91]. However, more recently, Wallace et al. [92] reported that there was no difference in the incidence of hepatic artery thrombosis and other potentially TACE-related complications such as biliary stricture and leaks after LT between patients with and without TACE. On the other hand, there is concern that widespread adhesions after RFA may increase complications after LT. However, Haas et al. [93] reported that there was no difference in postoperative mortality and morbidity between patients with and without RFA after propensity score matching.

There are some reports comparing the results of LT after two different locoregional therapies. Gyori et al. [94] compared the results after TACE and RFA as BT, and they reported that both had similar transplant rates, tumor responses, and post-transplant outcomes. Ward et al. [12], comparing hepatectomy and RFA for single tumors, reported that RFA was more common for local recurrence, but hepatectomy was more common for distant metastases. In their study, explanted histopathology of patients treated with RFA showed pathological CR in 85.7% of patients. Therefore, they concluded that LT candidates with solitary HCC <3 cm should be treated with ablation. In recent years, drug-eluting bead TACE (DEB-TACE), degradable starch microsphere TACE (DSM-TACE), Yttrium-90 radioembolization, and stereotactic body radiation therapy (SBRT), etc., have been used as bridging and downstaging modalities instead of conventional methods. [95,96,97,98,99,100,101,102]. DEB-TACE enables the maintenance of high concentrations of the drug in the tumor and fewer systemic adverse effects compared to conventional TACE by slowly and continuously releasing the anti-cancer drug locally. [103,104,105]. Frenette et al. [106] reported that both conventional TACE and DEB-TACE had similar frequencies of complete necrosis, post-transplant recurrence rates, and drop-outs from the waiting list. On the other hand, Nicolini et al. [107], comparing conventional TACE and DEB-TACE as pre-transplant locoregional therapy, reported that the recurrence-free survival rate was significantly better with DEB-TACE than with conventional TACE (61.5% vs. 87.4%, *p* = 0.0493). Although DEB-TACE had few systemic side effects compared to conventional TACE, Fidelman et al. [108] reported that there were many complications for patients with borderline hepatic reserve, and it was not safe to use as a pre-transplant locoregional therapy for Child-Pugh B patients. DSM-TACE using DSM, which is decomposed by blood amylase and has a short arterial embolization period in the body of up to 50 min, has been reported in cases of portal vein thrombosis [109,110,111]. Minici et al. [112,113] reported that DSM-TACE could be used safely and effectively as bridging or downstaging for patients with poor hepatic reserve (Child-Pugh score 8–9 points). Radioembolization using Yttrium-90 is reported to treat target lesions more completely for early intermediate stage HCC than conventional TACE [114]. Radiolabeled particles are trapped at the precapillary level within the tumor vasculature, thus limiting exposure to the surrounding normal parenchyma. This allows higher dose delivery than with external beam radiation therapy [115]. Gabr et al. [116], examining 207 cases of Yttrium-90 therapy as bridging or downstaging prior to LT, reported tumor response with complete necrosis (100%) in 94% of patients, extensive necrosis (51–99%) in 29%, and partial necrosis (<50%) in 26%, with a recurrence rate of 12% after LT. SBRT is an external beam radiation therapy method that precisely delivers a high dose of radiation to an extracranial target using either a single dose or a small number of high-dose fractions. The frequency of complete pathological necrosis of the tumor after SBRT varies from 13.3% to 61.5% [101,117,118]. Wong et al. [119] reported that pre-transplant SBRT had a higher frequency of complete pathological necrosis and a lower frequency of drop-outs from the waiting list compared to conventional bridging therapy.

In recent years, molecular-targeted therapies have been used alone or in combination with locoregional therapies as pre-transplant treatments [120,121]. Sorafenib is an oral multi-kinase inhibitor that has been effective in prolonging time-to-progression in patients with advanced cancer in two large phase III trials [122,123]. Recent evidence suggests that neo-angiogenic reactions are induced after TACE, which potentially enhances the tumor growth of untreated nodules or accelerates the development of de novo tumors [124]. Since sorafenib suppresses the growth of HCC by directly suppressing angiogenesis, it may be effective for bridging or downstaging when combined with TACE. However, at present, there are few reports that sorafenib was useful for bridging or downstaging before LT [121]. In addition to sorafenib, lenvatinib and regorafenib, which are also oral multi-kinase inhibitors, are used as therapeutic agents for advanced HCC. A meta-analysis showed that lenvatinib showed superior progression-free survival to sorafenib in patients with advanced HCC [125]. Regorafenib has also been shown to be useful as a second-line treatment after sorafenib [126]. Although it has been reported that successful down-staging was obtained with lenvatinib in a case report [127], no studies of lenvatinib and regorafenib as a neoadjuvant treatment before LT have been reported.

As explained above, there are various pre-transplant treatments for HCC used for bridging or downstaging. A complete understanding of the characteristics of each treatment is essential for selecting the optimal treatment, taking into account tumor burden, tumor site, and liver reserve.

## 6. Comprehensive Assessment of Transplantable Tumor for Assigning Priority for Organ Allocation

As mentioned earlier, T2 patients are given priority in the allocation system by UNOS in the USA. Even with the same T2 tumor, however, the risk of waiting list drop-out differs between cases with T2 at first presentation and cases after bridging or downstaging. Taking these facts into consideration, Mazzaferro [128] proposed the concept of a comprehensive assessment of transplantable tumor (TT), which involves classification into 8 subgroups considering the risk of drop-out based on tumor stage, suitability for locoregional therapies, and therapeutic effect, and gives priority to the group with the highest risk of drop-out. The 8 subclasses are as follows: TT0_C_, no residual tumor after curative treatment of HCC; TT0_L_, no residual tumor after locoregional embolic therapies for HCC; TT1, single HCC ≤2 cm; TT0 _NT_, no residual tumor after treatment of a non-transplantable HCC (successful downstaging); TT _FR_, transplantable HCC >T1 at first presentation or recurrent HCC >2 years after curative treatment; TT _UT_, transplantable HCC judged untreatable for reasons not captured by MELD (i.e., ascites); TT _PR_, partial response after complete bridge therapy in a transplantable tumor; TT _DR_, transplant eligibility after downstaging (sustained partial response) or recurrent HCC <2 years after curative treatment of any HCC. From TT0 _C_ to TT _DR_, the stage of the tumor progresses, and the higher priority for organ allocation is given in this order. This concept was included in the Italian Consensus-Based Approach to Organ Allocation in Liver Transplantation [129]. Sandro et al. [130] validated the allocation system using the TT staging system by classifying the 8 subgroups of TT into 3 groups, the high-risk group (TT _DR,_ TT _PR_), the intermediate-risk group (TT0 _NT_, TT _FR,_ TT _UT_), and the low-risk group (TT1, TT0 _L,_ TT0 _C_), in descending order of priority. They reported that the recurrence rate in the high-risk group was significantly lower when LT was performed within 2 months after staging (10% vs. 33% for <2 and >2 months, *p* = 0.006), which supported the validity of prioritization to the high-risk group by the TT staging system. However, the opposite result was obtained in the intermediate-risk group, and they concluded that further studies, including the definition of TT subgroups classified into the intermediate-risk group, are required.

## 7. Conclusions

The role of pre-transplant treatment, the characteristics of each treatment, and the results of LT after treatment were reviewed based on recent articles. The Milan criteria, which were proposed more than 20 years ago, are a very good morphological standard of LT for patients with HCC, and to date, pre-transplant treatment was aimed to keep the patients within these criteria for both patients within and outside the criteria. As a result, which patients need bridging, and the upper limit of tumor burden for downstaging have gradually become clear. It is expected that further studies with more effective bridging and downstaging therapies to obtain better survival and a lower recurrence rate of LT for HCC will be conducted.

## Figures and Tables

**Table 1 cancers-14-00396-t001:** Results of bridging locoregional therapies for HCC before LT in selected studies of the past 5 years.

Author	Country	Year	No. of Patients	Selection Criteria	Time Period on Waitlist to LT	Treatment Modality	Drop-out Rate	OS, RR, DFS, and RFS after LT
Lee [52]	USA	2017	121	MC (within MC 90.1%)	10.2 months	RFA	1-y 13.5%, 3-y 37.2%, 5-y 58.1%	RR 1-y 2.5%, 3-y 5.3%, 5-y 7.2%
Tan [33]	Singapore	2018	36	MC	291 (17–844) days	TACE, RFA	6-m 18.7%, 1-y 33.3%	3-y DFS 71%
Lee [39]	Korea	2020	T1 91 T2 54	OPTN T1, T2	NA	TACE, RFA, RT, Resection	NA	RFS T1 1-y 97.8%, 3-y 96.5%, 5-y 94% T2 1-y 94.8%, 3-y 92.1%, 90.7%
Na [53]	Korea	2016	53	MC	NA	TACE, RFA, PEI	Within→Beyond 24.5%	RFS 3-y 78.3%, 5-y 73.1%
Affonso [54]	Brazil	2019	136	MC	6.6 (0.60–30.47) months	DEB-TACE	33.8%	RFS 3-y 76.5%, 5-y 72.3%
Xing [55]	USA	2017	155	MC	5.92 (0.12–67.33) months	TAE, TACE, DEB-TACE, RE, RFA	28.3%	OS 3-y 85%, 5-y 72%
Agopian [38]	USA	2017	2854	MC	NA	TACE, RE, RFA, PEI, Resection, MW, etc.	NA	RFS 1-y 89%, 3-y 77%, 5-y 68%

OS: Overall survival; RR: Recurrence rate; DFS: Disease-free survival; RFS: Recurrence-free survival; LT: Liver transplantation; MC: Milan criteria; RFA: Radiofrequency ablation; TACE: Transarterial chemoembolization; RT: Radiation therapy; OPTN: Organ procurement and transplantation network; NA: Not available; PEI: Percutaneous ethanol injection; DEB: Drug-eluting beads; RE: Radioembolization; MW: Microwave ablation.

**Table 2 cancers-14-00396-t002:** Results of LT after downstaging of advanced HCC in selected studies of the past 5 years.

Authors	Country	Year	No. of Patients	Downstaging Criteria	Eligibility Criteria	Treatment Modality	Successful DS rate	OS, RR, DFS, and RFS after LT
Chapman [63]	USA	2017	210 (Within UCSF 35, Beyond UCSF 175)	MC	No extrahepatic lesions	TACE, RE, RFA	42.4%	RR Beyond UCSF 8.9%, within UCSF 5.6%
Mehta [58]	USA	2018	187	MC	UNOS DS inclusion criteria	TACE, RFA	83.4%	RR 10.1%
Massarollo [62]	Brazil	2016	85	MC	No extrahepatic lesions, no major vascular invasion	TACE	NA	OS 3-y 72.0%, 5-y 66.1%
Mehta [70]	USA	2020	543 (Within UNOS DS criteria 422, AC 121)	MC	No limitations	NA	NA	OS within UNOS DS 3-y 79.1%, AC 71.4% RR within UNOS DS 9.2%, AC 10.7%
Toso [72]	Switzerland	2019	29	TTV 115/AFP 400	No limit for size nor AFP	TACE, RFA, PEI, Resection, etc.	NA	DFS 5-y 74%
Ravaioli [64]	Italy	2019	122	MC	Single ≤8 cm, bifocal ≤5 cm, multiple ≤5 lesions ≤4 cm each with total tumor diameter ≤12 cm	TACE, RFA, Resection, PEI	68.4%	OS 5-y 63% RR 5-y 20.9%

OS: Overall survival; RR: Recurrence rate; DFS: Disease-free survival; RFS: Recurrence-free survival; LT: Liver transplantation; UCSF: University of California San Francisco; MC: Milan criteria; TACE: Transarterial chemoembolization; RE: Radioembolization; RFA: Radiofrequency ablation; NA: Not available; UNOS: United Network for Organ Sharing; DS: Downstaging; AC: All comers; TTV: Total tumor volume; AFP: Alpha-fetoprotein; PEI: Percutaneous ethanol injection.

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
