# Peer review of "Role of Pretransplant Treatments for Patients with Hepatocellular Carcinoma Waiting for Liver Transplantation"

_cancers, 2022, doi:10.3390/cancers14020396_

Round 1

Reviewer 1 Report

I read with much interest this review by Ogawa and Takada entitled "Role of pretransplant treatments for patients with hepatocellular carcinoma waiting for liver transplantation"

Neoadjuvant therapies settled as one of the most reliable index of tumor biology and aggressivenes, playing a crucial role on oncological outcomes of LT for HCC: as a consequence, a detailed review of the current bridging and downstaging strategies is particularly useful.

The whole paper is well organized and the different sections are well balanced.

I have some requests in order to improve the paper quality before its publication:

1) The Authors should include a recently published paper focusing on application of LI-RADS classification during preoperative workup of LT candidate in section 2: 10.1111/tri.13983

2) The recent paper from the Toronto group focusing on machine-learning algorithm for post-transplant HCC recurrence also deserves a mention in section 2: 10.1002/lt.26332.

3) The role of des-gamma-carboxy prothrombin/PIVKA as a biological marker of HCC aggressiveness, especially after neoadjuvant treatments, should also be reported: 10.5301/ijbm.5000276; 

4) I would also like some comments concerning the paramount issue of prioritization of LT candidates according to tumor biology, as recently speculated by Mazzaferro (10.1002/hep.28420) and further explored by Di Sandro et al. (10.3390/cancers11060741)

Best regards

Reviewer 2 Report

Very well written and comprehensive review on an important topic.

I have only some minor comments:

1) How were studies selected for Table 1? I would prefer a more comprehensive table....

2) I really liked the chapter on loco-regional therapies. However, i would add a brief comment on the non-superiority of TACE over bland embolization (TAE). In this regard, cite the recent meta-analysis PMID: 28588882)

3) Probably some brief comments on the cost-effectiveness of these treatments would be useful....

4) Among systemic treatments, please cite also the other approved agents: regorafenib (PMID: 31877664) and lenvatinib (PMID: 34017396)

Round 2

Reviewer 1 Report

Really well written review of the current bridging and downstaging strategies for HCC before LT.

Authors revisions improved the paper quality, that is now suitable  for publication